# Factors impacting resilience as a result of exposure to COVID-19: The ecological resilience model

**Anna Panzeri**[1], **Marco Bertamini**[1,2], **Sarah Butter**[3], **Liat Levita**[3], **Jilly Gibson-Miller**[3], **Giulio Vidotto**[1], **Richard P. Bentall**[3], **Kate Mary Bennett**[2]*

**1** Department of General Psychology, University of Padova, Padova, Italy, **2** Department of Psychological Science, University of Liverpool, Liverpool, United Kingdom, **3** University of Sheffield, Sheffield, England

* k.m.bennett@liverpool.ac.uk

## Abstract

Despite the severe psychological impact of the COVID-19 pandemic, some individuals do not develop high levels of psychological distress and can be termed resilient. Using the ecological resilience model, we examined factors promoting or hindering resilience in the COVID-19 pandemic. Of the 1034 participants (49.9±16.2 years; females 51.2%) from Italian general population, 70% displayed resilient outcomes and 30% reported moderate-severe anxiety and/or depression. A binary regression model revealed that factors promoting resilience were mostly psychological (e.g., trait resilience, conscientiousness) together with social distancing. Conversely, factors hindering resilience included COVID-19-anxiety, COVID-19-related PTSD symptoms, intolerance of uncertainty, loneliness, living with children, higher education, and living in regions where the virus was starting to spread. In conclusion, the ecological resilience model in the COVID-19 pandemic explained 64% of the variance and identified factors promoting or hindering resilient outcomes. Critically, these findings can inform psychological interventions supporting individuals by strengthening factors associated with resilience.

## Introduction

At the end of 2019, a new severe acute respiratory syndrome coronavirus 2 (SARS-CoV-2), hereafter referred to as COVID-19, was identified in Wuhan, China. By the end of January 2020, the virus had spread to Italy, the government declared a state of emergency, and the first confirmed cases occurred on the 21st of February and the first death the day after. Quarantine was imposed in Northern Italy on the 8th of March and on the 11th of March the whole country was placed in lockdown. Between then and the end of October more than 37000 deaths have been recorded at a rate of 620 per million.

A number of reports and papers have examined the negative effects of the pandemic on psychological wellbeing in the UK, USA, and China (for a recent meta-analysis see [1] and relatively less work on the impact in Italy. Most of the work to date has focused on the negative effects of COVID-19 on psychological wellbeing [2–4]. However, the psychological impact is

**Data Availability Statement:** Restrictions apply on the public availability of the data due to ethical reasons and license agreements to protect the privacy of participants. The data contains

potentially identifying and sensitive personal information. The data is available upon reasonable request to the Ethical Committee of the University of Padua (comitato.etico.area17@unipd.it).

**Funding:** The author(s) received no specific funding for this work.

**Competing interests:** The authors have declared that no competing interests exist.

likely to be more nuanced. Indeed, there may be people for whom the pandemic has not been detrimental. This study is focused on identifying the psychological, behavioural and community factors that can either promote or hinder resilient outcomes. Resilience can either be defined as a psychological trait that supports psychological wellbeing [5], or as the capacity for stability in wellbeing despite challenge [6] or as the ability to adapt and change or to *bounce back* in the face of challenge [7–10]. This current work takes into account all of these three perspectives (see also Kate Mary Bennett et al., 2019). Although we do not have pre-pandemic data, longitudinal studies have shown that the pandemic has impacted on a variety of outcomes (Cooke et al., 2020). Our work builds on Windle's [11] large-scale concept analysis and utilises her definition: 'Resilience is the process of negotiating, managing and adapting to significant sources of stress or trauma. Assets and resources within the individual, their life and environment facilitate this capacity for adaptation and *bouncing back* in the face of adversity. Across the life course, the experience of resilience will vary' (p. 163).

Central to all approaches to resilience is the requirement for a challenge or trauma. Thus, it differs from psychological wellbeing, quality of life or mental health: although they are influenced by the presence of challenges, the challenge itself is not a necessary component. Thus, resilience has been examined in a number of contexts: natural disasters [12, 13], bereavement [14], dementia care [15, 16], poverty [17], war [18] and other traumas [19, 20]. However, resilience has not been fully examined as yet in relationship to pandemics (see later for a discussion). Pandemics, by their nature, affect many thousands of people at a discrete point of time, so provide an opportunity to study resilience on a larger scale than would normally be the case. Following the concept analysis and definition developed by Windle [11], Windle and Bennett [21] developed the ecological model of resilience (Fig 1) which was originally developed in relationship to caregiving but has subsequently been used in other contexts, for example amongst older people living in poverty in Colombia [17], widowhood [10], chronic health conditions [22], and caregiving in relationship to end of life care [23] and dementia care [15].

The model takes as its starting point the idea that no person exists in isolation. Their lives and outcomes are influenced by the presence or absence of individual, community and societal resources. The model is not hierarchical, and the levels may impact on each other. To give a concrete example, access to respite care (a societal level resource) may influence family relationships (a community level resource) and also health (an individual level resource). The

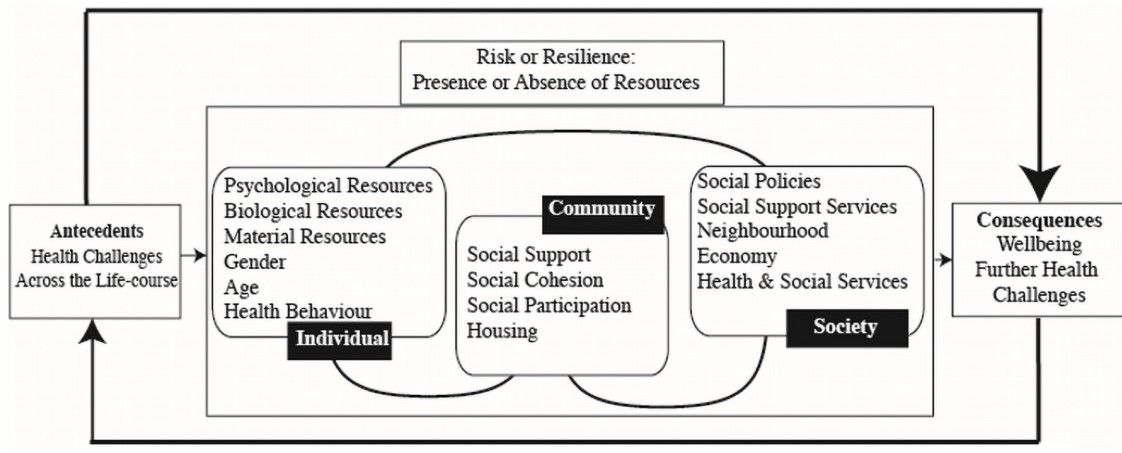

**Fig 1. The ecological model of resilience adapted from Windle and Bennett (2011).**

presence of resources facilitates resilience whereas the absence of resources hinders resilience and leads to compromised wellbeing or further challenges. It is important, therefore, to understand what resources facilitate or hinder resilience and which are amenable to change. In this conceptualisation resilience is considered a binary outcome (see [24] below for a discussion of measurement approaches). To our knowledge, to date, few studies focused on positive outcomes during the 2020 pandemic [25, 26] and even fewer on resilience [27, 28]. Those that do focused on trait resilience (see [10] for a discussion). None of these studies were specifically aimed at testing a well-established theoretic model such as the ecological resilience approach in a representative population sample.

## The current study

In this paper we applied the ecological model in light of the COVID-19 pandemic in the Italian context. The data analysed come from a cross-sectional study of the impact of COVID-19 on psychological and social experiences. In March 2020, a longitudinal, multi-country project was launched by the COVID-19 Psychological Research Consortium (C19PRC) in the UK [29]. The C19PRC study aims to assess and monitor the psychological, social, political, and economic impact of the pandemic in the general population, using longitudinal surveys and mixed-methods studies (more information is available from https://www.sheffield.ac.uk/psychology-consortium-covid19). Based on this survey, replications, partial replications and extensions have been conducted in other countries including the Republic of Ireland, Spain, Italy, the United Emirates and Saudi Arabia. The Italy COVID-19 study [30] collected data from July 13th to July 28th 2020, after the contagion peak (end of March) and after the end of the first strict national lockdown (May 18th).

As the Italian study was not designed specifically to examine the ecological model of resilience, some extra variables were added to the original model but there were also some omissions of resources not available to test (e.g., welfare and health service use, social cohesion). However, we were able to test all levels of the ecological model and, in a further development, we also examined a number of COVID-19 specific factors which we suggest might influence resilience which were included as an additional level in the model. This enabled us to examine not only COVID-19 as the necessary challenge to produce resilience but also to identify the influence of factors which only emerged as a consequence of the pandemic and the response to it. The additional COVID-19 level comprised COVID-19 specific anxiety, COVID-19 related traumatic stress, adherence to social distancing and hygiene, infection rates and a variable we termed exposure to COVID-19. The first two measured mental health in the light of COVID-19 (as distinct from other mental health variables), the second two behaviours which were introduced in response to COVID-19 with the intention of reducing the spread of the disease, and the third was the rate of infection at the time the survey was conducted. Finally, COVID-19 exposure is defined as having had or known someone who has had (or thinks they have had) COVID-19, similarly with respect to having been tested, or having known someone who has died (see methods for precise wording).

We addressed three research questions.

1. What proportion of the sample were resilient?

2. Did the ecological model explain resilient outcomes?

3. Which resources facilitated resilience, and which hindered resilience and to what extent?

## Materials and methods

This study is an analysis of one aspect of the data collected by the Italian C19PRC study [30]. This was an extension of the original C19PRC study launched in UK in March 2020 [29]. Data were collected during two weeks of July 2020, from the 13[th] to the 28[th]. At this point the population had already experienced the first peak (by the end of March 2020) and strict lockdown measures (since March 9[th]–until May 19[th]). The Italian study focused on four Italian regions selected because of their geographic location and their number of COVID-19 cases and deaths by the mid of July: Campania (4788 cases, 432 deaths), Lazio (8389 cases, 846 deaths), Lombardia (95118 cases, 16760 deaths) and Veneto (19432 cases, 2041 deaths). This study was approved by the Ethics Committee of the University of Padua (reference number: 3818) and the UK study by the Ethics Committee of the University of Sheffield (reference number: 033759).

### Participants

Participants were recruited by the survey company Qualtrics with stratified quota sampling to ensure that the socio-demographic characteristics of each regional population (age, gender, household income) were maintained. Participants were eligible to take part if they lived in one of the four selected Italian regions and if they were at least 18 years old. They were excluded if completion time was too fast ($<$ 11 minutes and 11 seconds) or too long ($>$ three days). The median completion time of the online survey was 41 minutes. For this paper, 4 participants were excluded because they did not complete all the relevant parts of the survey. A total of 1034 participants were included (529 females, 51%), the overall mean age was 49.9 (SD = 16.2 years). In line with the distribution of the population, most of participants lived in Lombardia (n = 390), followed by Lazio (N = 232), Campania (N = 226), and Veneto (N = 186), thus they were distributed among northern (55.71%) and southern regions (44.29%). Regarding education, a minority had only elementary or some secondary education (8.32%), nearly half had completed high school (50.87%) with a further 40.81% having attained a higher level of education. Married participants comprised 58.12% (N = 601) whereas never married participants comprised 26.60% of the sample (N = 275). Table 1 shows the characteristics of the sample. All respondents were informed about the aims of the study, conducted in agreement with the Code of Ethics of the Italian Association of Psychology [31], and provided informed consent.

### Measures

A cross sectional survey was administered. The measures included were in line with the original C19PRC-UK study to allow cross-cultural comparison. Here we describe only the measures tested in the model, for the full list and more details about the survey see Bruno et al. (2020) [30].

**Defining resilience, the dependent variable.** *Resilience* outcome was defined as a dichotomous variable resulting from the conjoint levels of depressive and anxious symptoms. Resilient individuals had low levels of depressive symptoms (PHQ-9 $<$10) [32] and low levels of anxiety symptoms (GAD-7 $<$10) [32, 33]. All other participants were defined as Non-resilient (PHQ $\geq$10 or GAD $\geq$10) (see Cosco et al., 2017 [24] for a review of measurement approaches).

*Depression* symptoms were measured with the Patient Health Questionnaire-9 (PHQ-9) [32], its 9 items reflect the DSM-IV diagnostic criteria for major depressive disorder. Participants indicated how often they had been bothered by each symptom over the past 2 weeks using a four-point Likert type scale from "not at all" (0) to "nearly every day" (3). Total scores range from 0 to 27, with scores of 5–9 = 'mild', 10–14 = 'moderate', and $\geq$15 = 'severe'

**Table 1. Characteristics of the sample.**

| Variables | Total sample | Females | Males |
|---|---|---|---|
| | **n = 1034** | **n = 529 (51%)** | **n = 505 (49%)** |
| **Age, m (SD)** | 49.9 (16.2) | 47 (15.4) | 53 (16.4) |
| *Northern regions*, n (%) | 576 (56%) | 299 (56.5%) | 277 (54.9%) |
| Lombardia, n (%) | 390 (38%) | 198 (37.4%) | 192 (38%) |
| Veneto, n (%) | 186 (18%) | 101 (19.1%) | 85 (16.8%) |
| *South-centre regions*, n (%) | 458 (44%) | 230 (43.5%) | 228 (45.1%) |
| Lazio, n (%) | 232 (22%) | 120 (22.7%) | 112 (22.2%) |
| Campania, n (%) | 226 (22%) | 110 (20.8%) | 116 (23%) |
| *Education* | | | |
| *Lower than degree* | 612 (59.2%) | 312 (59%) | 300 (59.4%) |
| Elementary schools, n (%) | 3 (0.003%) | 0 (0%) | 3 (0.59%) |
| Middle school, n (%) | 83 (8.03%) | 43 (8.13%) | 40 (7.92%) |
| Professional school, n (%) | 24 (2.32%) | 10 (1.89%) | 14 (2.772%) |
| High school, n (%) | 502 (48.5%) | 259 (48.96%) | 243 (48.119%) |
| *Higher than degree* | 422 (40.8%) | 217 (41%) | 205 (40.6%) |
| Bachelor's degree, n (%) | 97 (9.38%) | 48 (9.07%) | 49 (9.703%) |
| Master's degree, n (%) | 224 (21.7%) | 111 (20.98%) | 113 (22.376%) |
| Master, n (%) | 73 (7.06%) | 40 (7.56%) | 33 (6.535%) |
| Ph.D. | 28 (2.71%) | 18 (3.4%) | 10 (1.98%) |
| **In a relationship** | 403 (39%) | 313 (59.2%) | 318 (63%) |
| **Not in a relationship** | 631 (61%) | 216 (40.8%) | 187 (37%) |
| **Living alone** | 138 (13.3%) | 72 (13.6%) | 66 (13.1%) |
| **Not living alone** | 896 (86.7%) | 457 (86.4%) | 439 (86.9%) |
| **Number of minors living with** | | | |
| 0 | 677 (65.47%) | 323 (61.059%) | 354 (70.1%) |
| 1 | 182 (17.6%) | 111 (20.983%) | 71 (14.06%) |
| 2 | 134 (12.96%) | 71 (13.422%) | 63 (12.48%) |
| 3 | 33 (3.19%) | 18 (3.403%) | 15 (29.7%) |
| > = 4 | 8 (0.01%) | 6 (0.01%) | 2 (0.4%) |
| **House of property** | 700 (67.7%) | 323 (61.1%) | 377 (74.7%) |
| **House not of property** | 334 (32.3%) | 206 (38.9%) | 128 (25.3%) |
| **Income lower than 28.000 euro/y** | 428 (41.4%) | 247 (46.7%) | 181 (35.8%) |
| **Income higher than 28.000 euro/y** | 606 (58.6%) | 282 (53.3%) | 324 (64.2%) |
| **Employed** | 560 (45.8%) | 280 (52.9%) | 280 (55.4%) |
| **Unemployed** | 474 (54.2%) | 249 (47.1%) | 225 (44.6%) |
| **Precarious health (self)** | 167 (16.2%) | 69 (13%) | 98 (19.4%) |
| **Not precarious health (self)** | 867 (83.8%) | 460 (87%) | 407 (80.6%) |
| **Death Anxiety: Acceptance** | 12.54 (4.77) | 13.01 (4.83) | 12.05 (4.66) |
| **Death Anxiety: Externally generated** | 10.56 (3.86) | 11.2 (3.9) | 9.94 (3.73) |
| **Death Anxiety: Finality** | 11.34 (4.22) | 11.92 (4.23) | 10.72 (4.13) |
| **Death Anxiety: Thoughts** | 7.32 (3.08) | 7.55 (3.13) | 7.08 (3.01) |
| **BFI: Extraversion** | 5.79 (1.49) | 5.95 (1.50) | 5.64 (1.45) |
| **BFI: Agreeableness** | 6.72 (1.35) | 6.72 (1.36) | 6.73 (1.35) |
| **BFI: Conscientiousness** | 7.06 (1.43) | 7.05 (1.44) | 7.07 (1.41) |
| **BFI: Neuroticism** | 5.62 (1.64) | 5.75 (1.69) | 5.48 (1.57) |
| **BFI: Openness** | 6.91 (1.63) | 6.92 (1.68) | 6.90 (1.58) |
| **Loneliness** | 5.13 (1.67) | 5.34 (1.68) | 4.91 (1.63) |

*(Continued)*

**Table 1.** (Continued)

| Variables | Total sample | Females | Males |
|---|---|---|---|
| | **n = 1034** | **n = 529 (51%)** | **n = 505 (49%)** |
| **Self-esteem** | 4.58 (1.65) | 4.30 (1.76) | 4.87 (1.46) |
| **Trait resilience** | 19.77 (3.82) | 19.20 (3.89) | 20.38 (3.64) |
| **Intolerance of uncertainty** | 35.44 (9.22) | 36.16 (9.57) | 34.68 (8.79) |
| **With social support** | 699 (67.6%) | 351 (66.4%) | 348 (68.9%) |
| **Without social support** | 335 (32.4%) | 178 (33.6%) | 157 (31.1%) |
| **Belonging to neighbourhood** | 607 (58.7%) | 303 (57.3%) | 304 (60.2%) |
| **Not Belonging to neighbourhood** | 427 (41.3%) | 226 (42.7%) | 201 (39.8%) |
| **Belonging to wider neighbourhood** | 451 (43.6%) | 225 (42.5%) | 226 (44.8%) |
| **Not Belonging to wider neighbourhood** | 583 (56.4%) | 304 (57.5%) | 279 (55.2%) |
| **Neighbourhood trust** | 4.44 (1.70) | 4.46 (1.73) | 4.42 (1.66) |
| **Religious** | 813 (78.6%) | 427 (80.7%) | 386 (76.4%) |
| **Not religious** | 221 (21.4%) | 102 (19.3%) | 119 (23.6%) |
| **RT index Campania** | 0.93 | 0.93 | 0.93 |
| **RT index Lombardia** | 1.14 | 1.14 | 1.14 |
| **RT index Lazio** | 1.23 | 1.23 | 1.23 |
| **RT index Veneto** | 1.61 | 1.61 | 1.61 |
| **Hygiene behaviours** | 65 (9.17) | 65.52 (9.29) | 64.44 (9.01) |
| **Social Distancing** | 65.86 (9.61) | 66.6 (9.6) | 65.11 (9.57) |
| **COVID-19 Anxiety** | 53.6 (27.8) | 59.4 (26.3) | 47.5 (27.9) |
| **Trauma symptoms (ITQ)** | 12.60 (5.93) | 13.32 (5.99) | 11.84 (5.78) |
| **COVID-19 Exposure** | 467 (45.2%) | 237 (44.8%) | 230 (45.5%) |

*Note*. ITQ = international trauma questionnaire; BFI = Big Five Personality Inventory.

depression. The PHQ-9 psychometric properties are strong (for an overview, Kroenke et al., 2010 [34]) and the internal reliability of the scale was excellent in this study ($\alpha$ = 0.92).

*Anxiety* symptoms were assessed with the Generalized Anxiety Disorder Scale (GAD-7) [33], a 7-item tool about the frequency of anxiety symptoms (*e.g*., trouble relaxing, becoming easily annoyed or irritable). Participants indicated how often they had been bothered by each symptom over the past 2 weeks using a four-point Likert scale from 0 (not at all) to 3 (nearly every day). Scores range from 0 to 21 with a cut-off of 10 identifying the generalised anxiety disorder with good sensitivity (0.89) and specificity (0.82). The GAD-7 showed good psychometric properties [35], and the internal reliability of the scale was excellent ($\alpha$ = .94) in this study.

**Resources: The independent variables.** We next describe the independent variables as they appear in the levels of the model: Individual, Community, Societal and COVID-specific.

*Individual resources.* We first list the demographic variables included in the model. Age (continuous variable), gender (female = 1, male = 0), education (split as until/over high school), being in a relationship or not, living alone or not, number of underaged children living with, home owned or not, household income above 28.000 euro/year or not, employed or not. In addition, we included precarious health: the respondents were asked if they had any health condition (e.g., lung problems, heart disease, kidney disease, liver disease, conditions affecting the brain and nerves, diabetes, weakened immune system), and poor health was coded as present (1) or absent (0).

In addition to demographic variables, we included psychological variables.

*Death anxiety* (DA) was measured using the Death Anxiety Inventory (DAI) [36], which has 17 items across 4 dimensions. *DA acceptance* (6 items, α = .868) concerns the acceptance of the individual emotional dimension about death and its meaning. *DA externally generated* (4 items, α = 0.792) refers to situations or elements with an external reference to death in our cultural context (e.g., coffins, cemeteries) that can trigger unpleasant feelings. *DA finality* (4 items, α = 0.866) concerns a spiritual dimension wondering about mortality and the limits of human existence. *DA thoughts* (3 items, α = 0.845) is related to the frequent cognitive thoughts and concerns about own death. The response format is a 5-point Likert type scale from totally disagree (= 1) to totally agree (= 5).

*Personality traits* were assessed with the Big Five Inventory (BFI) [37], a 10-item measure with 5 scales, each for a personality trait: extraversion, agreeableness, consciousness, neuroticism, openness. Items are rated on 5-step scale from 1 = "disagree strongly" to 5 = "agree strongly", each scale ranges from 5 to 10. The BFI-10 has good reliability and validity.

*Intolerance of Uncertainty* was assessed with the Intolerance of Uncertainty Scale Short Form (IUS-12) [38, 39], with 12 items scored on a 5-point Likert scale from 1 "*not at all characteristics of me"* to 5 "*entirely characteristic of me"*. It includes both prospective cognitions ("unexpected events are negative and should be avoided") and inhibitory cognitions ("uncertainty leads to the inability to act") about uncertainty that are well described in a single dimension [40, 41]. The total score ranges from 12 to 60. The IUS-12 has good psychometric properties [42], and Cronbach's α was 0.904 in this study.

*Loneliness* was measured with the Loneliness Scale (LS) [43], a 3-item tool assessing the frequency of social (dis)connectedness (e.g., lacking companionship, isolation from others) in large-scaled population surveys. The response format is a 3-point scale from 1 = hardly ever to 3 = often) with a minimum of 3 and a maximum of 9. In this study, Cronbach's αwas 0.789.

*Self-esteem* was assessed with the Single-Item Self-esteem Scale (SISES) [44], respondents rated their agreement with one statement ("I have high self-esteem") using a 7-point Likert scale from 1 "not very true of me" to 7 "very true of me".

*Trait resilience* was measured with the Brief Resilience Scale (BRS) [45] with 6 items (e.g., "I tend to bounce back quickly after hard times") scored on a 5-point Likert scale from 1 "strongly disagree" to 5 "strongly agree", total scores range from 5 to 30. The BRS has good psychometric properties [46]. In this study, Cronbach's αwas 0.733.

*Community variables*. *Social support* was measured with a single question: "How often did you meet your family/friends in the last week?", with a 5-point response format from 0 = "not at all" to 4 = "every day". The subset without contacts was compared to the others.

*Neighbourhood connectedness* was assessed with three measures. Belongingness to neighbourhood was measured with a single question "How strongly do you feel you belong to your immediate neighbourhood?" scored on a 4-point response format from 1 "not at all" to 4 "very strongly". There were two specific questions on trust in neighbours asking about willingness to leave house keys to them, and whether they could be asked to buy groceries in case of need, scored on a 4-point scale from 1 "very uncomfortable" to 4 "very comfortable". Scores range from 2 to 8, with higher scores associated with higher trust and connectedness.

*Belongingness to wider neighbourhood* was assessed with two questions taken from the UK study [29]–"How much do you identify with (feel a part of, feel love toward, have concern for) your community?" and "How much would you say you feel involved when bad things happen to your community?"–scored on a 5-point format from 1 = "not at all" to 5 = "a lot". Scores were dichotomized according to the median.

**Societal resources.** *Religious*. Participants were asked whether they believed in any religion (Catholic, Jewish, Muslim, other = 1), or not (atheist, agnostic, none = 0) since being

religious in Italy has cultural as well as personal connotations and connects individuals to a broader societal framework [17, 47].

*Region.* In addition, we collapsed regions into northern and southern regions because of geographic, cultural, and historical reasons [48] and due to the number of COVID-19 cases– higher in northern regions. Veneto and Lombardia are northern (= 0) whilst Lazio and Campania were considered southern (= 1).

**COVID-19 specific variables.** We included a number of variables specifically related to COVID-19.

*The mean infection Reproduction number over Time* (RT index) is a measure of the COVID-19 diffusion in a given time despite the control measures. It was calculated for each region in the 14 days before to data collection [49]. Higher values indicate higher COVID-19 diffusion. The highest RT was in Veneto (1.61), followed by Lazio (1.23), Lombardia (1.14) and Campania with the lowest value (0.93).

*Exposure to COVID-19* was a dummy variable defined as the stressful experience (or not) of self, family or acquaintance being infected or tested for COVID-19 (whether the outcome was positive or negative: as there may be anticipatory anxiety), or a family member or acquaintance having died because of COVID-19, as well having been in self-isolation because of the (suspected) infection.

*Hygiene behaviours* were measured with 17 items based on the COM-B (Capability, Opportunity, Motivation-Behaviour, version 1) [50, 51] scored on a 5-point Likert scale ranging from 1 "strongly agree" to 5 "strongly disagree". In this study Cronbach's αwas 0.863.

*Social distance behaviours* were assessed with 17 statements according to the COM-B [50, 51] on a 5-point Likert scale from 1 "strongly agree" to 5 "strongly disagree". αwas 0.872 in this study.

*COVID-19 anxiety* was measured with a single item ("How anxious are you about the coronavirus COVID-19 pandemic?") on an electronic visual analogue scale to indicate the degree of anxiety from 0 "not at all anxious" on the left to 100 "extremely anxious" with 10-point increments. Higher scores reflected higher levels of COVID-19 related anxiety.

*Post-traumatic stress disorder* in relation to the COVID-19 experience was measured with the International Trauma Questionnaire (ITQ) [52] referring to the last month with the following instructions: "Listed below are some problems that people sometimes report in response to traumatic or stressful life events. Please read each question carefully in relation to your experience with the COVID-19 pandemic. Answer indicating how much you have been annoyed/bothered by that problem over the past 30 days". The ITQ has 6 items encompassing three clusters of symptoms of Re-experiencing, Avoidance, and Sense of Threat. A 5-point Likert scale from 0 (Not at all) to 4 (Extremely) generates scores ranging from 0 to 24. αwas 0.925 in this study.

## Statistical analysis

A multiple binary odds logistic regression was used to investigate if there was a relationship between psycho-social factors and resilience in the context of the challenging COVID-19 pandemic. The dependent binary variable was resilience, and the predictors were defined according to the theoretical background of the ecological framework of resilience. The regression blocks represented the different theoretical components of the model [21].

The individual component was entered in two blocks: demographics and precarious health were entered in Block 1; next the psychological variables were entered in Block 2. Block 3 included the community variables, namely social support, connectedness with close neighbours, connectedness wider neighbourhood. Block 4 consisted in societal variables: region

(north-south) and religion. Finally, Block 5 regarded the COVID-19-specific variables: hygiene practices, social distancing behaviours, regional infection rate in the last 14 weeks, anxiety related to COVID-19, COVID-19 related traumatic stress, and exposure to COVID-19 specific stressors. The regression assumptions were tested: i.e. linearity of the logit, absence of multi-collinearity, residuals.

## Results

### Characteristics of the sample

All the 1034 participants had lived through the pandemic and associated general stressors. Almost half of the sample ($n = 467$, 45.2%) were exposed to COVID-19 specific stressors, namely had symptoms and were tested for COVID-19 (13.6%), had been in self-isolation (15.1%), tested positive to COVID-19 (1.35%), knew someone close who was infected (23.3%), and/or lost someone because of COVID-19 (14.5%). Thus, the other half of the sample ($n = 567$, 54.8%) were not directly exposed to these COVID-19 stressors.

Considering the first question of this study, we wanted to define which proportion of the sample could be termed resilient according to their mental health. One fifth of the respondents showed moderate-severe levels of anxiety (GAD $\geq$10, n = 216, 20.9%, 95%CI = 0.185 0.235) with the majority showing lower anxiety levels (GAD $<$ 10, 818, 79.1%, 95%CI = 0.765–0.815). A quarter of participants reported moderate-severe depression levels (PHQ-9 $\geq$ 10) were 254 (24.6%, 95%CI = 0.220–0.273) whilst those with lower or absent depressive symptoms accounted for 780 (75.4%, 95%CI = 0.727–0.780). Table 2 shows the proportions. Considering together anxiety and depression, in the total sample, 8.7% (n 90) reported 'only' moderate-severe depression; 5% (n 52) displayed 'only' moderate-severe anxiety; and 15.9% (n 164) showed both moderate-severe anxiety and depression. Thus, seventy percent of participants met our criteria for Resilience (n = 728), whilst 30% were classified as non-resilient (n = 306).

Table 3 shows the descriptives of the resilient and not-resilient samples and the comparisons on the main variables through t tests for independent samples and chi-squares ($\chi 2$). Regarding the differences between the resilient and not-resilient groups shown in Table 3, in the non-resilient group there were more females ($\chi 2$ = 13.99, $p < .001$), individuals were slightly younger (t = 8.58; $p < .001$), and there were more people with "at-risk" conditions ($\chi 2$ = 5.423; $p < .02$) such as precarious health due to pre-existing health issues. Differently, the number of health-care workers did not differ across groups ($\chi 2$ = 0.391; $p >.05$). Literature has shown that these people already have a higher risk of developing psychological issues and the pandemic may represent a further stressor (e.g. diathesis stress model).

**Table 2. Anxiety and depression clinical cut-offs.**

| | Total sample | Females | Males |
|---|---|---|---|
| | **n = 1034** | **n = 529 (51%)** | **n = 505 (49%)** |
| *Depression (PHQ-9)* | | | |
| *Absent-Mild (0–9)* | 780 (75.4%) | 381 (72%) | 399 (79%) |
| *Moderate-Severe ($\geq$10)* | 254 (24.6%) | 148 (28%) | 106 (21%) |
| *Anxiety (GAD-7)* | | | |
| *Absent-Mild (0–9)* | 818 (79.1%) | 392 (74.1%) | 426 (84.3%) |
| *Moderate-Severe ($\geq$10)* | 216 (20.9%) | 137 (25.9%) | 79 (15.6%) |

Note: PHQ-9 = Patient Health Questionnaire 9 items; GAD-7 = Generalized Anxiety Disorder 7 items.

**Table 3. Descriptives of the resilient and not resilient sample.**

|  | Resilient | Not Resilient |  |
|---|---|---|---|
|  | n = 728 (70.4%) | n = 306 (29.6%) |  |
| Age, mean (sd) | 52.62 (15.65) | 43.52 (15.55) | t = 8.58; $p < .001$ |
| Sex |  |  |  |
| Female | 345 (47.4%) | 184 (60.1%) | $\chi2 = 13.99, p < .001$ |
| Male | 383 (52.6%%) | 122 (39.9%) |  |
| Precarious health (self) | 105 (14.4%) | 62 (20.3%) | $\chi2 = 5.423; p < .02$ |
| Not precarious health (self) | 623 (85.6%) | 244 (79.7%) |  |
| Health-care workers | 27 (3.7%) | 16 (5.2%) | $\chi2 = 0.391; p > .05$ |
| Depression (PHQ-9) | 3.72 (2.82) | 13.73 (5.12) | t = 32.22; $p < .001$ |
| Anxiety (GAD-7) | 3.25 (2.65) | 11.66 (4.81) | t = 28.79; $p < .001$ |
| PTSD (ITQ) |  |  |  |
| Presence of PTSD | 213 (29.3%) | 252 (82.4%) | $\chi2 = 245.42; p < .001$ |
| Continuous score | 10.39 (4.58) | 17.85 (5.47) | t = 20.97; $p < .001$ |

Note: PHQ-9 = Patient Health Questionnaire 9 items; GAD-7 = Generalized Anxiety Disorder 7 items.

## Regression

Regarding the second and third questions of this study, we used a regression model to test the ecological resilience model and to examine the factors associated with resilient outcomes.

Resilient (= 1) or non-resilient (= 0) was the outcome in the binary logistic regression.

Table 4 shows the blocks, and Table 5 shows the details of the final model.

In the first block (demographic variables), the significant predictors were female gender (b = -0.394, OR = 0.674), age (b = 0.029, OR = 1.030), number of children living with (b = -0.319, OR = 0.727), and precarious health of self (b = -0.930, OR = 0.394). The logistic pseudo-R squares ($R^2$) of the model were Cox and Snell $R^2$ = 0.104 and Nagelkerke $R^2$ = 0.148.

When adding the second block with the psychological variables, several were significant. These were: education (b = -0.531, OR = 0.588, $CI_{OR}$ = 0.385–0.894), number of children (b = -0.284, OR = 0.753, $CI_{OR}$ = 0.590–0.957), precarious health (b = -0.802, OR = 0.448, $CI_{OR}$ = 0.269–0.746), DAI acceptance (b = -0.089, OR = 0.915, $CI_{OR}$ = 0.845–0.989), DAI finality (b = 0.161, OR = 1.175, $CI_{OR}$ = 1.086–1.273), DAI thoughts (b = -0.231, OR = 0.793, $CI_{OR}$ = 0.726–0.866), Conscientiousness (b = 0.399, OR = 1.491, $CI_{OR}$ = 1.281–1.744), Neuroticism (b = -0.210, OR = 0.810, $CI_{OR}$ = 0.701–0.934), loneliness (b = -0.424, OR = 0.654, $CI_{OR}$ = 0.573–0.744), trait resilience (b = 0.154, OR = 1.167, $CI_{OR}$ = 1.091–1.250), and intolerance of uncertainty (b = -0.052, OR = 0.949, $CI_{OR}$ = 0.925–0.973). The model $R^2$ improved to Cox and Snell $R^2$ = 0.394 and Nagelkerke $R^2$ = 0.561.

No variables in Block 3, community, had a significant effect and the significant predictors remained the same as in the previous model (Table 4). The $R^2$ indexes did not improve: Cox and Snell $R^2$ = 0.395 and Nagelkerke $R^2$ = 0.562.

In Block 4, the society variables were added. Location was a significant predictor, southern regions were associated with non-resilient outcomes (b = -0.555, OR = 0.574, $CI_{OR}$ = 0.388–0.845). The other significant predictors remained the same as above. The $R^2$ indexes slightly improved: Cox and Snell $R^2$ = 0.4 and Nagelkerke $R^2$ = 0.569.

Finally, the COVID-19 variables were added in Block 5. Only social distancing behaviours was associated with a resilient outcome (b = 0.049, OR = 1.050, $CI_{OR}$ = 1.012–1.091). Those associated with non-resilient outcomes were COVID-19 anxiety (b = -0.010, OR = 0.990, $CI_{OR}$

**Table 4. Regression blocks.**

| | Block 1 | | | + Block 2 | | | + Block 3 | | | + Block 4 | | |
|---|---|---|---|---|---|---|---|---|---|---|---|---|
| | b | se | p | b | se | p | b | se | p | B | se | p |
| (Intercept) | -0.100 | 0.3643 | .785 | 2.671 | 1.398 | .056 | 2.784 | 1.453 | .055 | 2.918 | 1.466 | .047 |
| Female gender | **-0.395** | **0.1499** | **.008** | -0.252 | 0.206 | .221 | -0.245 | 0.207 | .235 | -0.264 | 0.207 | .202 |
| Age | **0.029** | **0.0059** | **< .001** | 0.013 | 0.007 | .076 | 0.013 | 0.008 | .092 | 0.014 | 0.008 | .065 |
| Education above high school | -0.180 | 0.1620 | .266 | **-0.531** | **0.214** | **.013** | **-0.522** | **0.216** | **.016** | **-0.499** | **0.218** | **.022** |
| In a relationship | 0.228 | 0.1840 | .216 | 0.040 | 0.243 | .871 | 0.003 | 0.245 | .989 | -0.037 | 0.246 | .882 |
| Living alone | 0.004 | 0.2447 | .987 | 0.359 | 0.311 | .250 | 0.344 | 0.314 | .273 | 0.259 | 0.315 | .410 |
| Number of minors | **-0.319** | **0.0897** | **< .001** | **-0.284** | **0.123** | **.021** | **-0.300** | **0.124** | **.016** | **-0.302** | **0.125** | **.016** |
| House property | -0.061 | 0.1602 | .703 | -0.056 | 0.206 | .785 | -0.062 | 0.208 | .764 | -0.077 | 0.209 | .712 |
| Income (above | 0.160 | 0.1709 | .347 | -0.088 | 0.220 | .689 | -0.075 | 0.223 | .736 | -0.173 | 0.228 | .448 |
| Employed | -0.230 | 0.1719 | .181 | -0.166 | 0.220 | .452 | -0.172 | 0.220 | .437 | -0.187 | 0.222 | .400 |
| Precarious health | **-0.930** | **0.2021** | **< .001** | **-0.802** | **0.259** | **.002** | **-0.794** | **0.262** | **.002** | **-0.783** | **0.263** | **.003** |
| DAI: acceptance | - | - | - | **-0.089** | **0.040** | **.026** | **-0.088** | **0.040** | **.029** | **-0.090** | **0.040** | **.025** |
| DAI: externally generated | - | - | - | -0.002 | 0.037 | .954 | -0.004 | 0.037 | .916 | 0.005 | 0.038 | .900 |
| DAI: finality | - | - | - | **0.161** | **0.040** | **< .001** | **0.161** | **0.041** | **< .001** | **0.157** | **0.041** | **< .001** |
| DAI: thoughts | - | - | - | **-0.231** | **0.044** | **< .001** | **-0.232** | **0.045** | **< .001** | **-0.234** | **0.045** | **< .001** |
| BFI: Extraversion | - | - | - | -0.024 | 0.071 | .733 | -0.026 | 0.071 | .711 | -0.041 | 0.072 | .575 |
| BFI: Agreeableness | - | - | - | 0.002 | 0.077 | .982 | <0.001 | 0.078 | .996 | 0.005 | 0.079 | .945 |
| BFI: Conscientiousness | - | - | - | **0.399** | **0.078** | **< .001** | **0.391** | **0.079** | **< .001** | **0.386** | **0.078** | **< .001** |
| BFI Neuroticism | - | - | - | **-0.210** | **0.073** | **.004** | **-0.205** | **0.073** | **.005** | **-0.217** | **0.074** | **.003** |
| BFI: Openness | - | - | - | -0.014 | 0.062 | .827 | -0.021 | 0.063 | .740 | 0.002 | 0.064 | .975 |
| Loneliness | - | - | - | **-0.424** | **0.066** | **< .001** | **-0.417** | **0.066** | **< .001** | **-0.433** | **0.067** | **< .001** |
| Self-esteem | - | - | - | -0.080 | 0.064 | .211 | -0.086 | 0.065 | .183 | -0.086 | 0.065 | .185 |
| Trait Resilience (BRS) | - | - | - | **0.154** | **0.034** | **< .001** | **0.158** | **0.034** | **< .001** | **0.164** | **0.035** | **< .001** |
| Intolerance of Uncertainty (IUS) | - | - | - | **-0.052** | **0.012** | **< .001** | **-0.053** | **0.013** | **< .001** | **-0.052** | **0.013** | **< .001** |
| Social support | - | - | - | - | - | - | -0.163 | 0.208 | .434 | -0.170 | 0.210 | .419 |
| Neighbourhood belongingness (yes) | - | - | - | - | - | - | 0.132 | 0.216 | .540 | 0.138 | 0.219 | .528 |
| Neighbours trust | - | - | - | - | - | - | -0.012 | 0.062 | .845 | 0.005 | 0.062 | .938 |
| Belongingness wider neighbourhood | - | - | - | - | - | - | 0.126 | 0.208 | .546 | 0.144 | 0.210 | .492 |
| Religious (yes) | - | - | - | - | - | - | - | - | - | 0.163 | 0.254 | .5201 |
| Southern regions | - | - | - | - | - | - | - | - | - | **-0.555** | **0.199** | **.005** |
| Cox & Snell R$^2$ | 0.104 | | | 0.394 | | | 0.395 | | | 0.400 | | |
| Nagelkerke R$^2$ | 0.148 | | | 0.561 | | | 0.562 | | | 0.569 | | |
| AIC: | 1164 | | | 785.6 | | | 791.7 | | | 787.1 | | |
| BIC: | 1218 | | | 904.2 | | | 930.1 | | | 935.3 | | |
| Residual Deviance, (df) | 1142 (1023) | | | 737.59 (1010) | | | 735.7 (1006) | | | 727.06 (1004) | | |

= 0.981–0.998) and COVID-19 related traumatic stress (b = -0.182, OR = 0.834, CI$_{OR}$ = 0.796–0.871). In this model precarious health and DAI acceptance were no longer significant. The final R$^2$ for the model were Cox and Snell R$^2$ = 0.451 and Nagelkerke R$^2$ = 0.642. The regression assumptions—linearity of the logit, absence of multicollinearity, residuals—were respected.

Fig 2 shows the final ecological model of resilience in the context of COVID-19.

Fig 3 shows the plot of the odds ratios of the final model.

**Table 5. Final regression model (Block 5).**

| | | + Block 4 | | | + Block 5 | | | Odds Ratio | | |
|---|---|---|---|---|---|---|---|---|---|---|
| | | b | se | p | b | se | p | 2.5% CI | value | 97.5% CI |
| | (Intercept) | 2.918 | 1.466 | 0.047 | 2.844 | 1.826 | 0.119 | 0.487 | 17.176 | 634.374 |
| Demographics | Female gender | -0.264 | 0.207 | 0.202 | -0.199 | 0.228 | 0.385 | 0.523 | 0.820 | 1.284 |
| | Age | 0.014 | 0.008 | 0.065 | 0.010 | 0.009 | 0.236 | 0.993 | 1.010 | 1.027 |
| | Education above high school | **-0.499** | **0.218** | **0.022** | **-0.659** | **0.238** | **0.006** | **0.324** | **0.518** | **0.822** |
| | In a relationship | -0.037 | 0.246 | 0.882 | -0.012 | 0.272 | 0.965 | 0.579 | 0.988 | 1.682 |
| | Living alone | 0.259 | 0.315 | 0.410 | 0.432 | 0.335 | 0.197 | 0.803 | 1.541 | 2.995 |
| | Number of minors | **-0.302** | **0.125** | **0.016** | **-0.270** | **0.137** | **0.049** | **0.583** | **0.763** | **0.997** |
| | House property | -0.077 | 0.209 | 0.712 | -0.033 | 0.227 | 0.886 | 0.620 | 0.968 | 1.509 |
| | Income (above | -0.173 | 0.228 | 0.448 | -0.180 | 0.247 | 0.467 | 0.514 | 0.836 | 1.355 |
| | Employed | -0.187 | 0.222 | 0.400 | 0.020 | 0.242 | 0.933 | 0.636 | 1.021 | 1.643 |
| | Precarious health | **-0.783** | **0.263** | **0.003** | -0.550 | 0.284 | 0.053 | 0.330 | 0.577 | 1.006 |
| Psychological | DAI: acceptance | **-0.090** | **0.040** | **0.025** | -0.077 | 0.043 | 0.075 | 0.850 | 0.926 | 1.007 |
| | DAI: externally generated | 0.005 | 0.038 | 0.900 | 0.066 | 0.042 | 0.112 | 0.985 | 1.068 | 1.160 |
| | DAI: finality | **0.157** | **0.041** | **<0.001** | **0.149** | **0.043** | **0.001** | **1.068** | **1.161** | **1.266** |
| | DAI: thoughts | **-0.234** | **0.045** | **<0.001** | **-0.167** | **0.050** | **0.001** | **0.766** | **0.846** | **0.933** |
| | BFI: Extraversion | -0.041 | 0.072 | 0.575 | 0.023 | 0.080 | 0.772 | 0.876 | 1.023 | 1.197 |
| | BFI: Agreeableness | 0.005 | 0.079 | 0.945 | -0.058 | 0.087 | 0.506 | 0.794 | 0.944 | 1.119 |
| | BFI: Conscientiousness | **0.386** | **0.078** | **<0.001** | **0.370** | **0.085** | **<0.001** | **1.230** | **1.448** | **1.714** |
| | BFI Neuroticism | **-0.217** | **0.074** | **0.003** | **-0.206** | **0.082** | **0.012** | **0.692** | **0.814** | **0.955** |
| | BFI: Openness | 0.002 | 0.064 | 0.975 | 0.052 | 0.071 | 0.466 | 0.916 | 1.053 | 1.211 |
| | Loneliness | **-0.433** | **0.067** | **<0.001** | **-0.381** | **0.074** | **<0.001** | **0.589** | **0.683** | **0.788** |
| | Self-esteem | -0.086 | 0.065 | 0.185 | -0.075 | 0.072 | 0.295 | 0.805 | 0.928 | 1.067 |
| | Trait resilience (BRS) | **0.164** | **0.035** | **<0.001** | **0.150** | **0.039** | **<0.001** | **1.078** | **1.161** | **1.255** |
| | Intolerance of Uncertainty | **-0.052** | **0.013** | **<0.001** | **-0.033** | **0.015** | **0.025** | **0.939** | **0.967** | **0.996** |
| Community | Social support | -0.170 | 0.210 | 0.419 | 0.0130 | 0.227 | 0.955 | 0.648 | 1.013 | 1.582 |
| | Neighbourhood belongingness (yes) | 0.138 | 0.219 | 0.528 | 0.220 | 0.237 | 0.354 | 0.783 | 1.246 | 1.985 |
| | Neighbours trust | 0.005 | 0.062 | 0.938 | 0.005 | 0.067 | 0.942 | 0.881 | 1.005 | 1.146 |
| | Belongingness wider neighbourhood | 0.144 | 0.210 | 0.492 | 0.257 | 0.227 | 0.258 | 0.829 | 1.293 | 2.021 |
| Society | Religious (yes) | 0.163 | 0.254 | 0.5201 | 0.365 | 0.275 | 0.184 | 0.839 | 1.441 | 2.470 |
| | Southern regions | **-0.555** | **0.199** | **0.005** | **-0.577** | **0.248** | **0.020** | **0.344** | **0.562** | **0.910** |
| COVID-19 exposure | Infection RT (mean 14 days) | - | - | - | -0.312 | 0.533 | 0.559 | 0.258 | 0.732 | 2.086 |
| | Hygiene behaviours | - | - | - | -0.032 | 0.020 | 0.102 | 0.931 | 0.968 | 1.006 |
| | Social distancing | - | - | - | **0.049** | **0.019** | **0.010** | **1.012** | **1.050** | **1.091** |
| | COVID-19 anxiety | - | - | - | **-0.010** | **0.005** | **0.0212** | **0.981** | **0.990** | **0.998** |
| | PTSD symptoms | - | - | - | **-0.182** | **0.023** | **<0.001** | **0.796** | **0.834** | **0.871** |
| | Exposure to COVID-19 | - | - | - | -0.341 | 0.213 | 0.109 | 0.468 | 0.711 | 1.078 |

**Note**. Block 4: Cox & Snell $R^2$ = 0.400; Nagelkerke $R^2$ = 0.569; AIC = 787.1; BIC = 935.3; Residual deviance: 727.06; df = 1004.

Final model: Cox & Snell $R^2$ = 0.451; Nagelkerke $R^2$ = 0.642; AIC 707.3; BIC 885.2; Residual deviance: 635.28, df = 998.

## Discussion

This study aimed to test three research questions.

First research question, what proportion of the sample were resilient? Our results indicated that 70% of the sample were resilient. Nevertheless, these data show an increase of anxiety and depression rates compared to previous data of 2019 about mental health in Italy with a

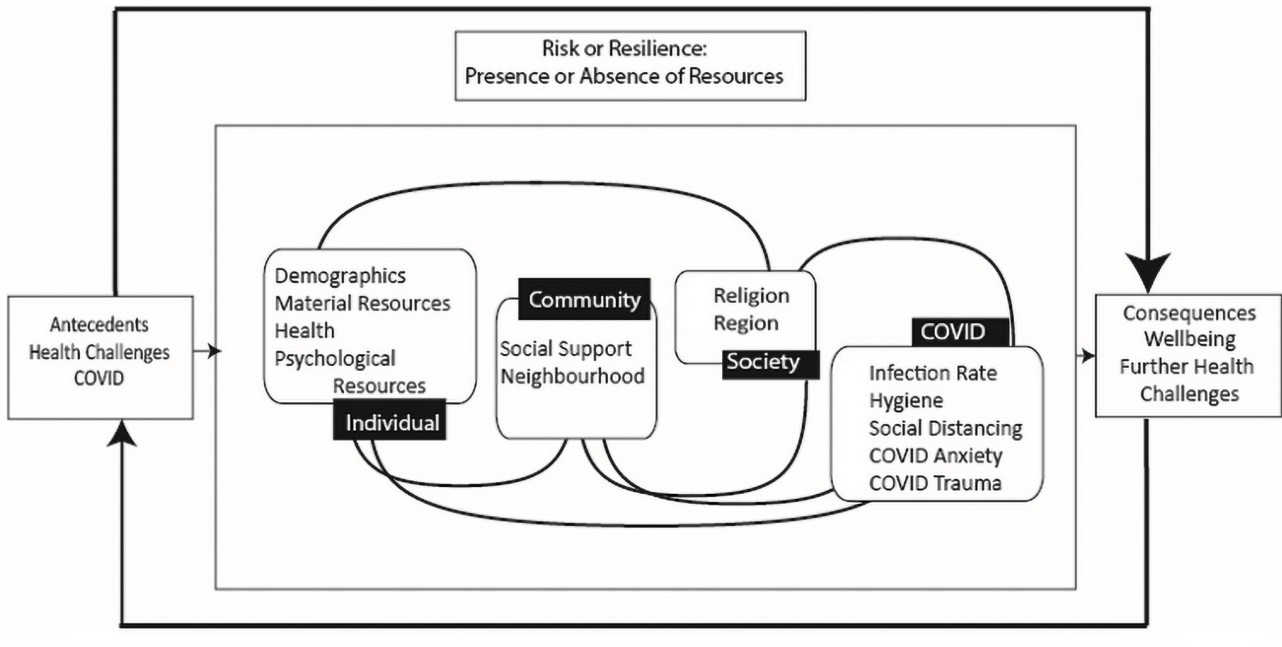

**Fig 2. The ecological model of resilience adapted to the context of COVID-19.** In this model we present those variables–individual, community, and societal–that were available in the Italian dataset to be tested. We have introduced specific variables in a COVID-19 level. Psychological variables which were significantly associated with resilience (+) and non-resilience (-) are: conscientiousness (+), neuroticism (-), death anxiety finality (+), intolerance of uncertainty (-), loneliness (-), and brief resilience scale (+).

prevalence of 7% for anxiety and 6% for depression [53]. These proportions of people showing caseness for generalised anxiety and depression were comparable to those obtained in the UK using the same measures during the early lockdown there [54]: the GAD prevalence was similar in Italy (20.9%) and UK (21.6%) but the PHQ-9 showed a slight increase in depression rates in Italy (24.6%) compared to UK (22.1%).

As a consequence, there is a need to understand which factors can promote or hinder resilience in order develop and implement evidence-based psychological interventions.

As a second research question, we tested whether the ecological model explained resilient outcomes. Resources at the individual and societal levels and in relationship to COVID-19-specific factors contributed to both resilient and non-resilient outcomes. However, none of the community resources were significant. Thus, the model was effective, and explained a significant amount of the variance.

Third research question, we asked which resources explain resilient and non-resilient outcomes. Significant variables associated with resilient outcomes included: Death Finality, conscientiousness, living in northern regions, and social distancing. On the other hand, higher education, children living in the household, Death Thoughts, neuroticism, loneliness, COVID-related anxiety and COVID-related traumatic stress were associated with non-resilient outcomes. These findings differ from those presented elsewhere of a tsunami of mental health problems [25, 55], or from those that focus on negative impacts [56, 57]. This study is one of the few focusing on resilience in the COVID-19 pandemic, examining the factors associated with the process of adaptation and development of positive outcomes [58, 59] and demonstrates that the majority of people were resilient during the first wave/lockdown of the

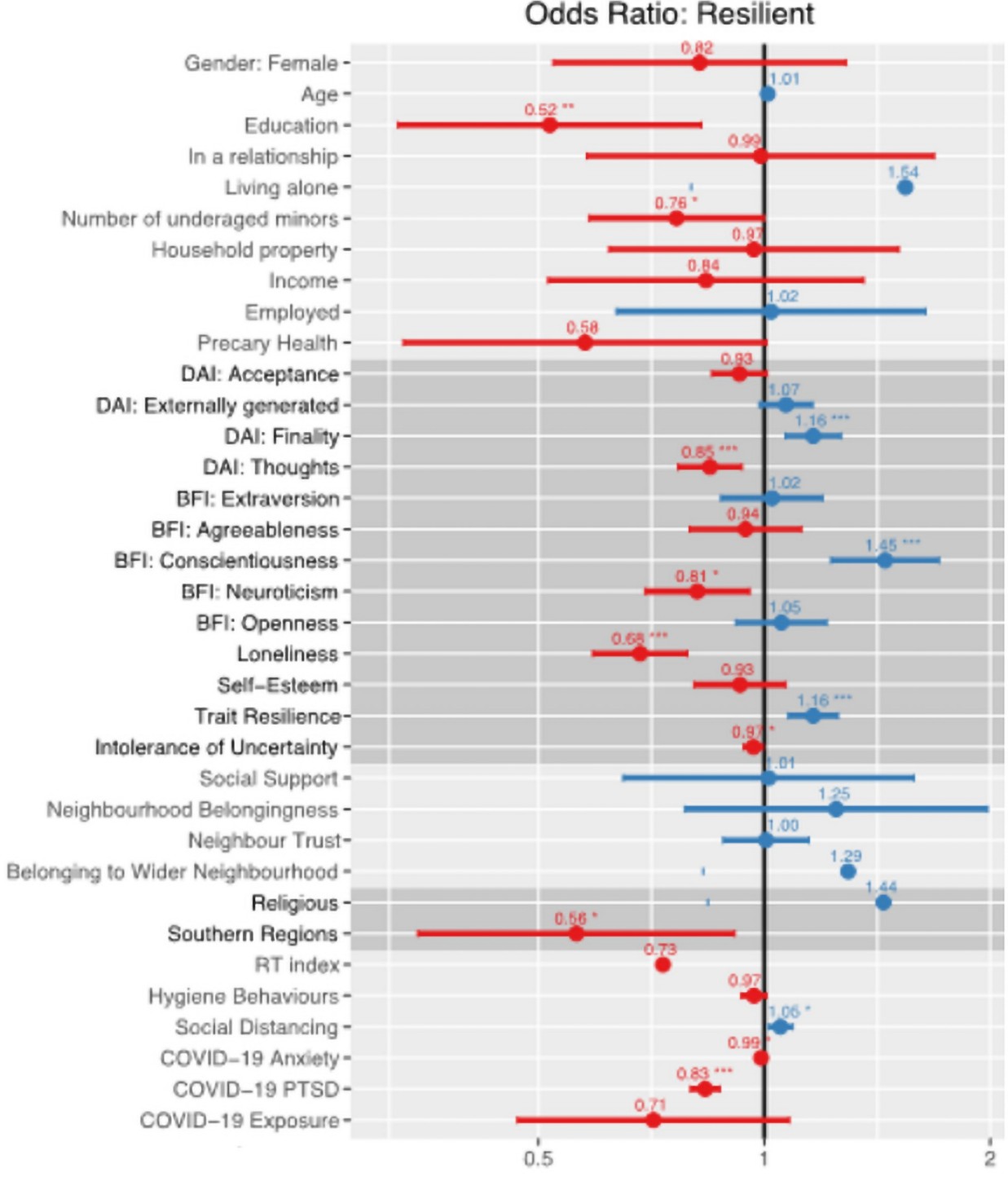

**Fig 3. Plot of the odds ratios with 95% confidence intervals based on the final regression model predicting resilient *vs.* not resilient outcomes.** On the left side (<1, red) there are the risk factors, on the right side (>1, blue) the protective factors. Different grey bands distinguish the blocks. Asterisks mark the significant predictors.

pandemic in Italy, as defined by not showing clinical levels of either anxiety or depression during the pandemic.

It is important in these challenging times to recognize that the majority of people take these adverse events in their stride. That is not to say that we should not be concerned about the 30% of people who are non-resilient and about other issues affecting participants, rather it is to ask what can we learn from the majority, and what mechanisms might explain why the majority are resilient?

One way of understanding the factors which contribute to resilience is within the ecological resilience framework [21]. Although the model was not originally designed for natural disasters or for pandemics, it allows us to consider a variety of factors which contribute to resilience. To reflect the COVID-19 pandemic and the nature of some of the specific COVID-related experiences we introduced a specific block into the model. As already noted, the community block did not contribute significantly to the model. This may be because the variables available did not measure this construct effectively, and in future research more sensitive measures would be beneficial. Similarly, whilst the societal level was significant this was only marginal. However, COVID-19-specific factors did significantly contribute to both resilient and non-resilient outcomes. Overall, the model explained 64% of the variance and was significantly effective in explaining resilient outcomes.

The third research question focused on which resources explained resilient and non-resilient outcomes: each level will be discussed in turn. We found that, as with other studies [60], the presence of children in the household had a detrimental effect. This is not surprising for a number of reasons: children had not been in school during the lockdown, parents would have experienced some degree of home schooling, and homes would have been more crowded for more of the time. Each of these factors may increase parental stress. Whilst normally higher education protects against a wide range of psychiatric issues [61], in this study, higher education levels were associated with non-resilience. At first glance one might find this surprising. However, Bennett et al. (2019) [10] in their longitudinal study of widowers found that the pattern of education associated with resilience was complex. One could speculate that those with higher education levels might seek out information about COVID-19 which was not optimistic or that those with higher education levels might be more likely to be home working. Amongst these risk factors, precarious health was also significantly associated with non-resilience–as one would expect, until the COVID-19 specific variables were included in the final model, suggesting that there is some degree of shared variance with the COVID-exposure variable.

Turning to the individual psychological variables, some variables related to personality traits such as neuroticism and conscientiousness were significant predictors and in the expected direction. One would expect those with high neuroticism (the disposition to be sensitive/nervous, as opposed to being confident) to be non-resilient, and those high on conscientiousness to be resilient–the latter probably through better self-regulation and the mechanism of engaging with COVID-related behaviours [62]. Loneliness, again as expected, was associated with non-resilient outcomes, possibly because loneliness can represent a trigger for anxiety and then depression [63, 64]. In addition, in our model high intolerance of uncertainty was associated with non-resilience. Uncertainty is one of the key features of the COVID-19 pandemic where the potential outcome is yet unknown. Our findings are in line with the uncertainty distress model [65], according to which the dispositional intolerance of uncertainty may act as a moderator strengthening the relationship between the perceived threat (e.g., COVID-19) and the uncertainty distress frequently experienced as worry and anxiety that could reach moderate-to-high anxiety levels, thus non-resilient states.

On the other hand, trait resilience as measured by the Brief Resilience Scale was, as one would expect, associated with state resilience confirming the results of other studies [66]

identifying trait resilience among the psychological features protecting a person against the negative stressors due to the pandemic [27, 67].

Three psychological variables are worthy of more detailed comment: Death Anxiety Finality; Death Anxiety Thought; and Self-esteem. Taking self-esteem first, one would have expected that high levels of self-esteem would have been associated with resilience. However, in preliminary analyses before we included death anxiety, self-esteem was associated with non-resilience. In the final model, self-esteem was not significant. This might be explicable within the framework of Terror Management Theory (TMT), which posits that individuals experiencing higher distress are those who need to rely the most on boosting their self-esteem and self-perceived value to improve their psychological health [68]. The DAI variables showed interesting patterns. Death acceptance and death thoughts (until the final model) were associated with not resilient outcomes. These can be thought of as internal and individual thoughts about one's own death, and awareness of one's own mortality. However, death anxiety finality was associated with resilience. This subscale reflects more impersonal and existential aspects of death. In the COVID-19 emergency people were frequently exposed to death related stimuli, as such these results suggest that thinking about death as a natural outcome to life is a protective factor.

Turning to the community variables, this block did not contribute significantly to the model. The variables available may not have measured this construct effectively. However, another possible interpretation is that in this difficult period of social distancing people were forced to rely on themselves, strengthening individualism and weakening their usual relationships with neighbourhood and their community. Similarly, whilst the societal level was significant this was only marginal. Among the society variables, living in a northern region (Veneto and Lombardia) was the only predictor of resilience. The northern regions were those hit harder by COVID-19, with a greater number of infections, deaths and with longer preventive measures. Thus, this result suggests a *steeling effect* [69]: being exposed to multiple stressors for a longer time is challenging but it also gives the opportunity to develop resilience.

Finally, the last block was introduced into the model to better reflect the COVID-19 pandemic and the nature of some of the specific COVID-related experiences. Engaging in social distancing behaviours were associated with resilient outcomes, although COVID-related hygiene behaviours were not. On the other hand, not surprisingly, PTSD symptoms related to COVID-19 and COVID-related anxiety (to a lesser extent) were associated with non-resilient outcomes, suggesting that the COVID-19 pandemic could be considered respectively as a traumatic event and an impersonal threat [70].

In the light of these findings, it is important to note that state resilience can be predicted by some stable psychological characteristics as the ones included in the psychological block (e.g., personality traits, dispositional intolerance of uncertainty). Studying the stable psychological characteristics favouring state resilience can improve the efficiency of psychological interventions.

Interestingly, these findings are in line with current literature suggesting that during the stressful times of a pandemic, some individuals are more resilient, and some are more fragile than others [71]. As in a Spanish study [72], in the Italian non-resilient group there were more females, more people with pre-exiting health issues, and slightly younger individuals. According to literature (e.g., Bonanno & Diminich, 2013) [73], age may have a curvilinear relationship with resilience, with the younger and elder individuals at higher risk for psychological issues [74–76]. Thus, particular attention should be dedicated to individuals with unfavourable conditions and the already vulnerable groups, as great elderlies [63, 77], patients with pre-existing medical diseases–e.g., cardiac, oncological [26, 75, 78–82], and health-care professionals [83,

84] for which the COVID-19 may have increased the risk of developing dangerous consequences and detrimental effects on both physical and mental health.

Our findings do need to be viewed in light of the following limitations. First, the cross-sectional nature of the data allows to observe associations among variables but not to infer causal relationships. In the future, longitudinal data would be useful to examine causal relationships between resources in the ecological model and resilient outcomes both with respect to the COVID-19 pandemic and more generally. The inclusions of more targeted variables such as social support, social participation, health and welfare service use and social benefits which were not available in this study would be valuable. It would also be useful to consider what interventions might be developed to support resilience both with respect to COVID-19 but also for other future pandemics or natural disasters. Second, as the study was online, only digitally-literate people were reached. In addition, the data was not collected originally to test the ecological model in Italy but was adapted to it–with some limitations about the scales not yet validated in the current language. Given the importance of using psychometrically sound validated tools, future studies will provide their Italian validations [85–88]. However, later waves of the Italian and UK surveys have been developed to include a more direct, longitudinal test of this model. Also, due to theoretical reasons of the resilience model, a logistic regression was chosen whereas a continuous one may have provided a better fit. Nevertheless, the model proved useful in explaining the resilient outcomes ($R^2 = 64\%$) and highlights factors associated with resilience in the time of COVID-19. The strongest contributing factors in identifying resilient and non-resilient outcomes were the psychological variables.

This was the first study to examine the ecological model of resilience at the time COVID-19. One of the key features of this study is the use of the ecological approach to resilience, considering not only the individual factors but contextualising them in a broader environment including the socio-economic conditions, the community, and the broader society [21]. Considering only the individual level is not sufficient to understand the complex factors which lead to resilience. One needs to take account of community and societal factors. A second original feature is the inclusion of COVID-19 specific factors in a separate block, identifying their unique contribution to resilience. The method of analysis demonstrated the impact of each level (e.g., individual) and how they vary in relationship to each other. This highlights the need for a comprehensive, ecological view to understand resilience more fully. Another methodological strength of this study consisted of adopting a stratified quota sampling methodology. A fair representation of the sociodemographic characteristics of the population adds to the validity and generalisability of results. This feature distinguishes the present research from studies conducted in the general population with convenience sampling (often through social media) and with limited representativeness of the referral population [28].

Despite a large number of studies highlighting the psychological burden related to COVID-19 (for a review, Xiong et al., 2020) [25], the present study is one of the few focusing on favourable psychological outcomes in the times of COVID-19 [89]. Although some investigated resilience [28] and the impact of positive psychological traits [27, 62], our study overcomes their limitations in two ways. We utilised a robust stratified sampling methodology which reflects the population. And we started from a theoretical background, the ecological approach, which drives the research aims and the analyses, and it includes both trait and state resilience [10].

In conclusion, more than 70% of the sample were classified as resilient despite the adverse ongoing pandemic. Significant variables associated with resilient outcomes included: death anxiety (finality), conscientiousness, living in northern regions, and social distancing. On the other hand, higher education, children living in the household, death thoughts, neuroticism, loneliness, COVID-related anxiety and COVID-related traumatic stress were associated with non-resilient outcomes. Notably, the ecological model was effective in explaining resilient

outcomes during the COVID-19 pandemic. Factors facilitating and hindering resilience were identified. Individual, societal and COVID-19 specific factors, but not community factors, were associated with resilient and non-resilient outcomes.

Regarding the clinical implications of this model, these findings may be useful for implementing and planning clinical interventions with benefit at both the individual and societal level [90–92]. Our findings suggest that it is important to provide psychological support–for individuals with children and women. Importantly, psychological interventions could strengthen in particular the factors at the individual psychological level that are associated with resilience––such as reducing the effects of intolerance of uncertainty by providing clear and easy to understand information about the risks of the pandemic, promoting heathy behaviours as social distancing, and reducing COVID-19 related anxiety and managing PTSD symptoms [93].

In conclusion, the present study provided a better understanding about psychological adjustment to a pandemic event and allowed to distinguish which groups of the general population are more at risk of facing higher levels of psychological distress and which factors contribute to resilience. This knowledge could be useful in the prevention and treatment of analogous stressful and traumatic conditions to promote the individuals' psychological health within the wider societal framework.

## Author Contributions

**Conceptualization:** Anna Panzeri, Marco Bertamini, Sarah Butter, Liat Levita, Jilly Gibson-Miller, Giulio Vidotto, Richard P. Bentall, Kate Mary Bennett.

**Data curation:** Anna Panzeri.

**Formal analysis:** Anna Panzeri.

**Investigation:** Marco Bertamini, Sarah Butter, Liat Levita, Giulio Vidotto, Richard P. Bentall, Kate Mary Bennett.

**Methodology:** Anna Panzeri, Marco Bertamini, Giulio Vidotto, Kate Mary Bennett.

**Software:** Anna Panzeri.

**Supervision:** Marco Bertamini, Jilly Gibson-Miller, Giulio Vidotto, Richard P. Bentall, Kate Mary Bennett.

**Writing – original draft:** Anna Panzeri, Marco Bertamini, Kate Mary Bennett.

**Writing – review & editing:** Anna Panzeri, Marco Bertamini, Sarah Butter, Liat Levita, Jilly Gibson-Miller, Giulio Vidotto, Richard P. Bentall, Kate Mary Bennett.

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
