## [Decision Letter · Decision Letter 0]

21 Jun 2021

PONE-D-21-09431

Factors Impacting Resilience as a Result of Exposure to COVID-19:

the Ecological Resilience Model

PLOS ONE

Dear Dr. Panzeri,

Thank you for submitting your manuscript to PLOS ONE. After careful consideration, we feel that it has merit but does not fully meet PLOS ONE’s publication criteria as it currently stands. Therefore, we invite you to submit a revised version of the manuscript that addresses the points raised during the review process.

We look forward to receiving your revised manuscript.

Kind regards,

Giovanni Ottoboni

Academic Editor

PLOS ONE

Journal Requirements:

Reviewers' comments:

Reviewer's Responses to Questions

**Comments to the Author**

1. Is the manuscript technically sound, and do the data support the conclusions?

Reviewer #1: Yes

Reviewer #2: Yes

2. Has the statistical analysis been performed appropriately and rigorously? 

Reviewer #1: Yes

Reviewer #2: Yes

3. Have the authors made all data underlying the findings in their manuscript fully available?

Reviewer #1: Yes

Reviewer #2: No

4. Is the manuscript presented in an intelligible fashion and written in standard English?

Reviewer #1: Yes

Reviewer #2: Yes

5. Review Comments to the Author

Reviewer #1: Very good work, no comments from my side and I recomend the paper for publication .However , the clinical implications of this model needed to be added in discussion

Limitations needed to specfic to this study..and Limitations regarding scale should also be mentioned in discussion.

Reviewer #2: # Overall:

The paper describes an observational study about the psychological impact of Covid-19 on a representative sample of the Italian general population. The paper is very interesting, well written and with a good theoretical background. The methods and the statistics are sound and robust. The results provide interesting information across all the levels of the ecological resilience model. Discussions are interesting, they present important topics with a fresh style, also disclosing interesting perspectives for the clinical and research field.

I add just some suggestions to improve the paper.

# 1: Since the literature about Covid-19 impact has rapidly increased, I suggest to check if other updated studies have been published. The authors could consider including them in the introduction and the discussion sections – with some comments if necessary.

Methods:

# 2: A question for the authors: why a logistic regression was preferred and not a continuous one? Probably, the models with continuous outcomes would have provided better fit. Some theoretical reasons for this choice are provided but the methodological issues should be mentioned discussed.

# 3: Page 11, line 221: The IUS has more items than the 12 used in the study. The 12 item version is called IUS-R. Please correct the information about the IUS tool and provide the correct Italian reference.

# 4: page 13, line 288: it would be useful for the readers to add the exact wording of the question about Covid-19 anxiety.

Results:

# 5: Please consider to add a table with the descriptives of the resilient and not resilient sample, also including the main psychological variables (e.g., Gad, Phq, Itq, …). Indeed, this information is interesting because (maybe) in the non resilient group there are more people with “at-risk” conditions, such as patients with pre-existing health issues, elderlies, health-care workers, young people, or females. Literature has shown that these people already have an higher risk of developing psychological issues and the pandemic may represent a further stressor (e.g. diathesis stress model). This is also an interesting point for the discussions.

Discussion:

# 6: Page 22, line 436: Please elaborate more about the role of the significant predictors that also already emerged in covid-19 literature, as the intolerance of uncertainty (Freeston et al., 2021).

# 7: In the discussion section, the authors may compare this study results with those of similar studies (e.g., DOI: 10.1017/SJP.2021.7; DOI: 10.1080/20008198.2020.1871555).

# 8: Just a few questions about the impact of this study:

(a) How the authors expect that resilient and not resilient individuals will behave over time? Is resilience predictive of future psychological health?

(b) If resilience is constructed over time by facing difficult situations, do we expect that non-resilient individuals will become resilient?

(c) Is it possible that ‘state resilience’ is predicted by some stable psychological characteristics - as the ones in the psychological block?

6. PLOS authors have the option to publish the peer review history of their article (what does this mean?). If published, this will include your full peer review and any attached files.

Reviewer #1: **Yes: **Sheikh Shoib

Reviewer #2: **Yes: **Anna Parola

---

## [Author Response · Author response to Decision Letter 0]

27 Jul 2021

RESPONSE TO REVIEWERS 

Reviewers' Comments to the Authors

1. Is the manuscript technically sound, and do the data support the conclusions?

Reviewer #1: Yes

Reviewer #2: Yes

2. Has the statistical analysis been performed appropriately and rigorously?

Reviewer #1: Yes

Reviewer #2: Yes

3. Have the authors made all data underlying the findings in their manuscript fully available?

Reviewer #1: Yes

Reviewer #2: No

Authors’ Answer: As in line with the journal guidelines, we added the data availability statement at the end of the manuscript.

 4. Is the manuscript presented in an intelligible fashion and written in standard English?

Reviewer #1: Yes

Reviewer #2: Yes

5. Review Comments to the Author

Reviewer #1:

Very good work, no comments from my side and I recomend the paper for publication. However , the clinical implications of this model needed to be added in discussion

Limitations needed to specfic to this study..and Limitations regarding scale should also be mentioned in discussion.

Answer: Thank you for appreciating the manuscript and for the comment.

According to your comment, we deepened both the clinical implications and the limitations section as follows:

“Our findings suggest that it is important to provide psychological support– for individuals with children and women. Importantly, psychological interventions could strengthen in particular the factors at the individual psychological level that are associated with resilience –– such as reducing the effects of intolerance of uncertainty by providing clear and easy to understand information about the risks of the pandemic, promoting heathy behaviours as social distancing, and reducing COVID-19 related anxiety and managing PTSD symptoms.”

Reviewer #2:

# Overall:

The paper describes an observational study about the psychological impact of Covid-19 on a representative sample of the Italian general population. The paper is very interesting, well written and with a good theoretical background. The methods and the statistics are sound and robust. The results provide interesting information across all the levels of the ecological resilience model. Discussions are interesting, they present important topics with a fresh style, also disclosing interesting perspectives for the clinical and research field. I add just some suggestions to improve the paper.

# Answer: Thank you so much for appreciating our manuscript and for providing useful comments. Below we provided a point to point explanation of how we tried to address your comments.

# 1 Comment: Since the literature about Covid-19 impact has rapidly increased, I suggest to check if other updated studies have been published. The authors could consider including them in the introduction and the discussion sections – with some comments if necessary.

#1 answer: Thank you, we updated the manuscript with recent studies about the psychological impact of Covid. 

Methods:

# 2 Comment: A question for the authors: why a logistic regression was preferred and not a continuous one? Probably, the models with continuous outcomes would have provided better fit. Some theoretical reasons for this choice are provided but the methodological issues should be mentioned discussed.

#2 answer: As the reviewer said, the logistic model was preferred because of theoretical reason relying on the ecological resilience model that conceptualizes resilience as binary state: an individual may be resilient or not resilient - depending on her/his levels of anxiety and depression. For sure, a continuous dependent variable would have provided more precise estimates and a higher pseudo-R^2. In this regard, we added a comment in the limitation section:

“Also, due to theoretical reasons of the resilience model, a logistic regression was chosen whereas a continuous one may have provided a better fit.”

# 3 Comment: Page 11, line 221: The IUS has more items than the 12 used in the study. The 12 item version is called IUS-R. Please correct the information about the IUS tool and provide the correct Italian reference.

#3 answer: Thanks for noticing this error. We corrected it.

# 4 Comment: page 13, line 288: it would be useful for the readers to add the exact wording of the question about Covid-19 anxiety.

#4 answer: Thanks, we added it:

“COVID-19 anxiety was measured with a single item (“How anxious are you about the coronavirus COVID-19 pandemic?”) on an electronic visual analogue scale to indicate the degree of anxiety from 0 “not at all anxious” on the left to 100 “extremely anxious” with 10 point increments. Higher scores reflected higher levels of COVID-19 related anxiety.”

Results:

# 5 Comment: Please consider to add a table with the descriptives of the resilient and not resilient sample, also including the main psychological variables (e.g., Gad, Phq, Itq, …). Indeed, this information is interesting because (maybe) in the non resilient group there are more people with “at-risk” conditions, such as patients with pre-existing health issues, elderlies, health-care workers, young people, or females. Literature has shown that these people already have an higher risk of developing psychological issues and the pandemic may represent a further stressor (e.g. diathesis stress model). This is also an interesting point for the discussions.

#5 answer: thank you. We added this table (Table 3) in the paper and we also commented about this point in the results and the discussion section – also integrating updated literature - as follows:

Results:

“Regarding the differences between the resilient and not-resilient groups, in the non-resilient group there are more females (χ2 = 13.99, p < .001), individuals are slightly younger (t = 8.58; p < .001), and there are more people with “at-risk” conditions (χ2 = 5.423; p < .02) such as precarious health due to pre-existing health issues. Differently, the number of health-care workers did not differ across groups (χ2 = 0.391; p > .05).”

Discussion:

“Interestingly, these findings are in line with current literature suggesting that during the stressful times of a pandemic, some parts of society are more fragile than others. In the non-resilient group there were more females, slightly younger individuals, and more people with pre-exiting health issues. Thus, particular attention should be dedicated to individuals with unfavourable conditions and the already vulnerable groups, as great elderlies (Webb, 2020; Van As et al., 2021; Panzeri et al., 2021a), patients with pre-existing medical diseases – e.g., cardiac, oncological (Ben Gal et al., 2020; Rossi et al., 2021; Ayubi et al., 2021; Rossi Ferrario et al., 2021; Panzeri et al., 2020), and health-care professionals (De Kock et al., 2021; Panzeri et al., 2021b) for which the COVID-19 may have increased the risk of developing dangerous consequences and detrimental effects on both physical and mental health. As a consequence, there is a need to understand which factors can promote or hinder resilience in order develop and implement evidence-based psychological interventions.”

Discussion:

# 6 Comment: Page 22, line 436: Please elaborate more about the role of the significant predictors that also already emerged in covid-19 literature, as the intolerance of uncertainty (Freeston et al., 2021).

#6 answer: thanks for this comment. We added some sentences to deepen the role of some predictors as suggested.

“Turning to the individual psychological variables, some variables related to personality traits such as neuroticism and conscientiousness were significant predictors and in the expected direction. One would expect those with high neuroticism (the disposition to be sensitive/nervous, as opposed to being confident) to be non-resilient, and those high on conscientiousness to be resilient – the latter probably through better self-regulation and the mechanism of engaging with COVID-related behaviours (Zager Kocjan et al., 2021). Loneliness, again as expected, was associated with non-resilient outcomes, possibly because loneliness can represent a trigger for anxiety and then depression (Van As et al., 2021; A. Rossi et al. 2020).”

“Uncertainty is one of the key features of the COVID-19 pandemic where the potential outcome is yet unknown. Our findings are in line with the uncertainty distress model (Freeston et al., 2021), according to which the dispositional intolerance of uncertainty may act as a moderator strengthening the relationship between the perceived threat (e.g., Covid-19) and the uncertainty distress frequently experienced as worry and anxiety that could reach moderate-to-high anxiety levels, thus non-resilient states.”

“On the other hand, trait resilience as measured by the Brief Resilience Scale was, as one would expect, associated with state resilience confirming the results of other studies (Osimo et al., 2021) identifying trait resilience among the psychological features protecting a person against the negative stressors due to the pandemic (Morales-Vives et al., 2020; Barzilay et al., 2020).”

# 7 Comment: In the discussion section, the authors may compare this study results with those of similar studies (e.g., DOI: 10.1017/SJP.2021.7; DOI: 10.1080/20008198.2020.1871555).

#7 answer: Thank you. In the discussion section, we added a part about these studies and current literature about the topic.

“Interestingly, these findings are in line with current literature suggesting that during the stressful times of a pandemic, some individuals are more resilient, and some are more fragile than others (Valiente et al., 2021a). As in a Spanish study (Valiente et al., 2021b), in the Italian non-resilient group there were more females, more people with pre-exiting health issues, and slightly younger individuals. According to literature (e.g., Bonanno & Diminich, 2013), age may have a curvilinear relationship with resilience, with the younger and elder individuals at higher risk for psychological issues (Parola et al., 2020; Panzeri et al., 2021a). Thus, particular attention should be dedicated to individuals with unfavourable conditions and the already vulnerable groups, as great elderlies (Webb, 2020; Van As et al., 2021), patients with pre-existing medical diseases – e.g., cardiac, oncological (Rossi et al., 2021; Ayubi et al., 2021; Rossi Ferrario et al., 2021; Ben Gal et al., 2020; Panzeri et al., 2020), and health-care professionals (De Kock et al., 2021; Panzeri et al., 2021b) for which the COVID-19 may have increased the risk of developing dangerous consequences and detrimental effects on both physical and mental health.”

# 8 Comment: Just a few questions about the impact of this study:

(a) How the authors expect that resilient and not resilient individuals will behave over time? Is resilience predictive of future psychological health?

(b) If resilience is constructed over time by facing difficult situations, do we expect that non-resilient individuals will become resilient?

(c) Is it possible that ‘state resilience’ is predicted by some stable psychological characteristics - as the ones in the psychological block?

#8 Answer: 

Thank you for this interesting comment. Below we provide a whole response including all the three points.

a) The resilient or non resilient trajectories are an interesting and complicated topic at the same time. Some longitudinal studies (Valiente et al. 2021) showed that individuals can be distinguished in 4 main different categories that tap the pattern of responses after traumatic events (Bonanno, 2004; Galatzer-Levy et al., 2018): a) Recovered (i.e., presence of distress at T1, absence at T2); b) Resilient (i.e., absence of distress at T1 and T2); c) Sustained distress (i.e., presence of distress at T1 and T2); and d) Delayed distress (i.e., absence of distress at T1, presence at T2).

b) It is possible that non-resilient individuals will become resilient but it is not certain. Indeed, a stressor is a necessary but not sufficient condition in order to build resilience. Beyond experiencing a stressful situation, resilience is built upon various factors about we commented above (e.g., coping, personality, etc).

The fact that the pandemic may represent the ‘first’ strong stressor for the young individuals would explain their higher levels of anxiety and depression.

Also, the current pandemic may not be the first stressing condition that most of individuals are facing, thus we expect that a stressor can have a steeling effect for most of the individuals, but literature showed that a part of individuals maintain non-resilient outcomes over time.

c) Finally, it is possible that ‘state resilience’ is predicted by some stable psychological characteristics - as the ones in the psychological block. We added some comments in this regard in the discussion section:

“In the light of these findings, it is important to note that state resilience can be predicted by some stable psychological characteristics as the ones included in the psychological block (e.g., personality traits, dispositional intolerance of uncertainty). Studying the stable psychological characteristics favouring state resilience can improve the efficiency of psychological interventions.”

---

## [Editor Report · Decision Letter 1]

29 Jul 2021

Factors Impacting Resilience as a Result of Exposure to COVID-19:

the Ecological Resilience Model

PONE-D-21-09431R1

Dear Dr. Panzeri,

We’re pleased to inform you that your manuscript has been judged scientifically suitable for publication and will be formally accepted for publication once it meets all outstanding technical requirements.

Kind regards,

Giovanni Ottoboni

Academic Editor

PLOS ONE
---

## [Editor Report · Acceptance letter]

9 Aug 2021

PONE-D-21-09431R1 

Factors Impacting Resilience as a Result of Exposure to COVID-19: the Ecological Resilience Model 

Dear Dr. Panzeri:

I'm pleased to inform you that your manuscript has been deemed suitable for publication in PLOS ONE. Congratulations! Your manuscript is now with our production department. 

Kind regards, 

on behalf of

Dr. Giovanni Ottoboni 

Academic Editor

PLOS ONE